# Tilorone and Cridanimod Protect Mice and Show Antiviral Activity in Rats despite Absence of the Interferon-Inducing Effect in Rats

**DOI:** 10.3390/ph15050617

**Published:** 2022-05-17

**Authors:** Viktoriya Keyer, Laura Syzdykova, Gulzat Zauatbayeva, Aigerim Zhulikeyeva, Yerlan Ramanculov, Alexandr V. Shustov, Zarina Shulgau

**Affiliations:** National Center for Biotechnology, Korgalzhin hwy 13/5, Nur-Sultan 010000, Kazakhstan; keer@biocenter.kz (V.K.); syzdykova@biocenter.kz (L.S.); zauatbaeva@biocenter.kz (G.Z.); zhulikeeva@biocenter.kz (A.Z.); ramanculov@biocenter.kz (Y.R.); shulgau@biocenter.kz (Z.S.)

**Keywords:** Tilorone, Cridanimod, interferon inducing agent, small animal models, Venezuelan equine encephalitis virus, antiviral state

## Abstract

The synthetic compounds, Tilorone and Cridanimod, have the antiviral activity which initially had been ascribed to the capacity to induce interferon. Both drugs induce interferon in mice but not in humans. This study investigates whether these compounds have the antiviral activity in mice and rats since rats more closely resemble the human response. Viral-infection models were created in CD-1 mice and Wistar rats. Three strains of Venezuelan equine encephalitis virus were tested for the performance in these models. One virus strain is the molecularly cloned attenuated vaccine. The second strain has major virulence determinants converted to the wild-type state which are present in virulent strains. The third virus has wild-type virulence determinants, and in addition, is engineered to express green fluorescent protein. Experimentally infected animals received Tilorone or Cridanimod, and their treatment was equivalent to the pharmacopoeia-recomended human treatment regimen. Tilorone and Cridanimod show the antiviral activity in mice and rats and protect the mice from death. In rats, both drugs diminish the viremia. These drugs do not induce interferon-alpha or interferon-beta in rats. The presented observations allow postulating the existence of an interferon-independent and species-independent mechanism of action.

## 1. Introduction

The continuing SARS-CoV-2 pandemic shattered societies across the globe. In fact, epidemiologists have predicted an upcoming major event on the global epidemiological scene because regional epidemics of deadly or severely debilitating diseases happen regularly [1]. Since the beginning of the twenty-first century, there were epidemics of severe acute respiratory syndrome (SARS) in 2002–2004, avian influenza A (H5N1 and then H5Nx) in 2009 and continuing years, human pandemic influenza H1N1 and N7N9, Middle East respiratory syndrome (MERS) in 2012–2013, Ebola hemorrhagic fever in 2013–2016 and Zika fever in 2015–2016 (reviewed in [1]). The biological nature of pathogens that will cause epidemics in the future can only be predicted with some levels of accuracy, and often there are no specific remedies at the time of the epidemic onset. This means that nonspecific, broad-activity treatment and prophylaxis options will be used for a prompt response. One paradigm to deal with a novel pathogen is stimulating innate immunity [2]. The innate immunity comprises a variety of defensive pathways of which the cellular defense against viruses regulated by type-I interferons (IFNs) is of critical importance [3]. IFNs are cytokines which are naturally produced by infected cells and they act on both infected and uninfected cells. This action results in activating numerous defensive genes and making the intracellular environment not permissive to viral replication [3]. There are also synthetic and natural compounds which, upon administration into the body, result in the increased production of IFNs, and eventually the establishment of non-permissive conditions to viruses (antiviral state) [4,5].

The subject of this study is the antiviral action of Tilorone and Cridanimod, which are low molecular weight (low-Mw) compounds that, in different countries, had or have approvals for clinical use. Tilorone and Cridanimod are in use as antivirals with a broad-spectrum activity in Kazakhstan, Russia, China, and other countries in the larger part of Eurasia. Both compounds can be given via different routes of administration, and they work for both prophylaxis and treatment.

Tilorone and Cridanimod are the oldest-known representatives in a class of low-Mw IFN-inducers. Tilorone and Cridanimod in experiments on mice, or in specific mouse cells, recognize the intracellular signaling protein stimulator of interferon genes (STING), resulting in the STING-TBK1-IRF3 pathway activation and synthesis of type I IFNs (subtypes IFN-α and IFN-β) [5].

Attempts to clinically apply Tilorone and Cridanimod led to an unresolved controversy which is presented in this paragraph. The Tilorone and Cridanimod structures were selected long ago on the basis of their ability to induce IFNs in mice [5,6], anticipating that the drugs would also induce IFNs in humans. Later, it was found that neither substance evidently increases circulating IFNs in patients [7,8]. Nevertheless, Tilorone and Cridanimod remain in the group “IFNs inducers” in national pharmacopoeias in the countries mentioned above [9]. Despite their ability to induce IFN-α/β in humans was not proven, Tilorone and Cridanimod have accumulated evidence of clinical efficacy as antivirals. Papers reporting both in vitro antiviral properties and the clinical efficacy are numerous (some are cited in the Discussion section). Research in western countries also confirms that these drugs have antiviral activity, including against SARS-CoV-2, MERS-CoV, and Chikungunya virus [7,10,11].

We have a scientific interest in studying whether these drugs have an IFN-α/β-independent mechanism of action, because if such mechanism exists, its study will resolve the above controversy. For this work, models of viral infections in mice and rats were developed, and the antiviral activity of Tilorone and Cridanimod was tested in both species. Experiments in rats are of special interest for the goal of this work because rats were previously shown to profile similarly to the human response to low-Mw IFN-inducers [8]. Experiments in mice had been designed to compare the virulence.

Strains of the Venezuelan equine encephalitis virus (VEEV) were used in this study. VEEV belongs to the genus Alphavirus, family Togaviridae. VEEV is a small-enveloped virus with a positive-sense RNA-genome [12]. This virus caused large epizootics in horses which sometimes were accompanied with human outbreaks in North, Central and South America. Different VEEV strains naturally circulate in wild rodents [13,14], and we expected that VEEV is suitable for developing a viral-infection model in laboratory rats. The only existing attenuated vaccine strain, TC-83, was derived from the virulent strain, Trinidad donkey (TrD), using multiple sequential passages. Comparing sequenced genomes of the vaccine and parental strains allowed the identification of the major virulence determinants in the VEEV genome [15]. Three VEEV variants were developed for this study by genetically engineering a molecular clone of the VEEV genome corresponding to the vaccine strain. The engineered viruses differ by the type of the major virulence determinants (vaccine or wild-type), and one variant expresses green fluorescent protein (GFP).

## 2. Results

### 2.1. Viruses

Three viruses designated cTC-83, cTC-83/TrD and cTC-83/TrD-GFP are variants of VEEV and were constructed with the intention of producing viruses differing in virulence. One virus produces GFP to conveniently track the infection. Differences between genomes of the viruses are shown in Figure 1.

The cTC-83 virus in this study represents the vaccine virus TC-83. cTC-83 was rescued from a molecular clone (the prefix “c” stands for “molecularly Cloned”). The TC-83 genome is the same as the Genbank entry EEVNSPEPB. The cTC-83/TrD virus differs by two nucleotide substitutions as compared with the parental cTC-83 as detailed in Materials and Methods. To create cTC-83/TrD, one nucleotide substitution was introduced in the 5′-untranslated region (5′UTR) and one in E2 gene. Both substitutions affected major virulence determinants, increased homology to the wild-type virulent strain Trinidad donkey (TrD), and were created to increase the virulence. The cTC-83/TrD-GFP virus is a derivative of cTC-83/TrD devised to produce GFP during the intracellular replication. A new copy of the VEEV subgenomic promoter (SP) was introduced in the viral genome as depicted in Figure 1. A GFP gene was placed downstream of the first SP copy and the second SP copy was left to control genes encoding VEEV structural proteins.

The three viruses were rescued by transfecting molecular infectious clones (MICs) DNA into BHK-21 cell cultures as described in Materials and Methods. Replication of all viruses induced the cytopathic effect (CPE) in infected cells resulting in cell death; the majority of the cells were dead by day 3 (photographs presented in Appendix A). This CPE enabled using a plaque test to determine titers. Titers of all the three viruses produced in cell cultures exceeded 10^8^ PFU/mL in samples collected at 72 h post-transfection (hpt). The virus cTC-83/TrD-GFP induced bright GFP fluorescence in the infected culture (Appendix A). The observed characters of the rescued viruses such as the high titers, CPE and prominent GFP expression confirm that the MICs are correctly assembled.

### 2.2. Viremia in Mice and Rats

Viremia profiles in mice for the viruses cTC-83, cTC-83/TrD and cTC-83/TrD-GFP are shown in Figure 2. The viremias in animals are different despite all viruses achieving high and similar titers in cell cultures. The vaccine-derived virus cTC-83 caused only low-titer viremia in mice. In contrast, the virus cTC-83/TrD, which has the two major virulence determinants, converted to the wild-type state and induced viremia four orders of magnitude higher (cTC-83/TrD vs. cTC-83 at any time point from 24 h post infection (hpi) to 96 hpi, *p* = 0.0001). Unexpectedly, the virus cTC-83/TrD-GFP, which differs from cTC-83/TrD, by having an insertion of a foreign gene induced a different viremia profile. The GFP-expressing virus cTC-83/TrD-GFP induced rapidly rising titers during the first day; however, at the end of the second day, the cTC-83/TrD-GFP viremia showed a sudden drop, whereas at that time, cTC-83/TrD sustained high titers. The described drop in titers was not an artifact caused by a loss of the GFP gene because during titrations, counts of GFP-positive foci and virus plaques gave the same titers. All viremias in mice declined below the detection limit at 120 hpi. Mice infected with the either virus strain showed no overt signs of disease during this short-course experiment.

Viremia profiles in rats are shown in Figure 3. In rats infected with cTC-83, the viremia was detected in only a fraction of animals and only in samples collected at 24 hpi. All samples collected at other times were negative in the plaque test. The titers in the cTC-83-viremic samples at 24 hpi were quite low, giving only 1–3 plaques in the first (1:10) dilution. The other two viruses carrying wild-type virulence determinants produced viremia in all infected rats, but the titers were ~100 times lower than in mice. Duration of the detectable viremia in rats was short. The rats remained visibly healthy.

### 2.3. Effects of Tilorone and Cridanimod on Viremia

In this experiment, animals infected with viruses received regular doses of Tilorone, Cridanimod, or a placebo, and the effect on titers was measured. The substances were delivered intragastrically (IG). Only two viruses, cTC-83/TrD and cTC-83/TrD-GFP, were used in this experiment because the cTC-83 virus appeared to be incapable of producing high titers in rats and mice and the cTC-83 viremia lasted for just one day in rats. The used animal treatment regimens for mice and rats were equivalent to the human treatment protocol.

Both Tilorone and Cridanimod show an effect on viremia in mice but the time profile and magnitude of the influence are different for the two drugs (Figure 4a,b). Daily doses of Tilorone reduce the titers at any time point during the observation. Tilorone has the highest effect at 24 hpi, whose time corresponds to the peak of viremia in the untreated animal control. Differences in titers at this time point were statistically significant between the control and Tilorone-treated animals (cTC-83/TrD vs. Control *p* = 0.0047; and cTC-83/TrD-GFP vs. Control *p* = 0.0014). Cridanimod has shown an even more pronounced effect at 24 hpi. Titers in the Cridanimod-treated mice were even lower than in the Tilorone-treated mice at 24 hpi (for cTC-83/TrD, Cridanimod vs. Tilorone *p* = 0.0022; and for cTC-83/TrD-GFP, Cridanimod vs. Tilorone *p* = 0.0005); however, the viremia suddenly peaks at 36 hpi in the Cridanimod-treated group but not in the group for Tilorone. This 36-hpi peak of viremia was observed for both cTC-83/TrD and cTC-83/TrD-GFP. This 36-hpi peak is short in duration. The titers in the Cridanimod-treated mice are again lower than in the control and Tilorone-treated mice at 48 hpi (Figure 4a,b). The observations of the near-zero viremia at 24 hpi and the peak at 36-hpi in the Cridanimod-treated group were reproducible in three biological replications of this experiment.

The cTC-83/TrD virus produces higher titers and longer viremia than cTC-83/TrD-GFP (compare the two control groups presented in Figure 4a,b). Receiving treatment with Tilorone or Cridanimod diminishes the viremia, and this curative effect is more pronounced with the more virulent virus. No deaths in mice from the infection or drugs’ toxicity were observed in this experiment.

The effect of Tilorone and Cridanimod on the viremia in rats was studied with the two viruses cTC-83/TrD and cTC-83/TrD-GFP because viremia of the attenuated cTC-83 virus was not detectable in all infected rats. Figure 5 presents viremia profiles in the control and treated rats. Tilorone diminishes titers in rats at every time point of observation in a fashion similar to what was observed in mice. This curative effect is more pronounced with the more virulent virus cTC-83/TrD. For example, the difference in the titers in the control and Tilorone-treated rats at 24 hpi is statistically significant for the cTC-83/TrD virus (*p* = 0.0003). Impressively, Cridanimod reduces the titers even more then Tilorone, below the limit of detection at this time point (*p* < 0.0001); however, the effect of Cridanimod is more complex, as a peak of viremia comes later in the Cridanimod-treated rats (Figure 5a,b). The timing for the peak is 36-hpi. The low viremia at 24 hpi (which is the time of maximum viremia in the control and Tilorone groups) and the aappearance of the 36-hpi peak in rats receiving Cridanimod was reproduced in three biological replicates of this experiment.

The presented data show that both drugs exhibit the curative effect in two rodent species, of which rats were previously shown to respond to the treatment with low-Mw IFN-inducers, which is more similar to the human response [8]. The two drugs, Tilorone and Cridanimod, show different dynamics of the antiviral action. The Cridanimod’s profile allows postulating the faster development of the effect, but with a shorter duration, compared with Tilorone.

### 2.4. Differences in Lethality in Untreated Mice and Rats

We found that only the virus cTC-83/TrD is capable of killing adult mice upon peripheral injection. Neither the attenuated cTC-83 virus, nor the virus based on cTC-83/TrD with a heterologous insert (cTC-83/TrD-GFP) killed adult mice. All viruses caused disease, and symptoms gradually increased for about six days. The common signs were apathy, rugged fur, and a hunched pose. A fraction of the mice infected with cTC-83/TrD developed paralysis. The highest daily mortality among the cTC-83/TrD-infected mice was on the sixth day (Figure 6a). A median lethal dose (LD_50_) for the cTC-83/TrD virus in adult mice was 10 PFU (data from the experiment to calculate the cTC-83/TrD virus’ LD_50_ for mice are presented in Appendix A). All mice infected with other strains cTC-83 or cTC-83/TrD-GFP recovered by the end of the 10 day observation period, and at that time, they looked healthy.

Rats infected with cTC-83/TrD showed lethargy and decreased consumption of feed. No deaths from the infection were seen in rats for any virus (Figure 6b).

### 2.5. Protection from Lethal Infection in Drugs-Treated Mice

Three groups of mice were injected SC with 10 LD_50_ of the virulent cTC-83/TrD virus. Tilorone, Cridanimod, or a placebo were given IG just before the virus injection and subsequently every day. The influence of drugs on survival is shown in Figure 7. Both Tilorone and Cridanimod show the statistically significant protection. On the last day of the experiment, 60% of mice survived in the Tilorone-treated group and the same 60% survived in the Cridanimod-treated group. There were no survivors in the placebo group. The Mantel–Cox test confirms the statistical significance of the survival (Tilorone vs. Control *p* = 0.0006; Cridanimod vs. Control *p* = 0.0002). The drug-treated survivors showed no visible signs of continuing disease.

### 2.6. Tilorone and Cridanimod Induce IFN-α/β in Mice but Not in Rats

Experiments in mice and rats were conducted to possibly link the antiviral effect of Tilorone and Cridanimod to IFN-α/β induction. The results show that administering Tilorone or Cridanimod to mice increases IFN-α and IFN-β with different dynamics (Figure 8). The induction of IFN-α is faster compared with IFN-β. For both drugs, the IFN-β concentration peaks lag behind the peaks of IFN-α, as is evident from comparing curves in Figure 8a,b. Tilorone showed only one concentration peak, despite the fact that mice received three doses at intervals; however, Cridanimod showed two peaks for both subtypes, IFN-α and IFN-β. During this bimodal induction, IFN-α is also faster than IFN-β (Figure 8). Possible explanations for the bimodality in the time profiles of IFN-α and IFN-β induced by Cridanimod are discussed in the Discussion section.

We did not observe the induction of IFN-α/β by either drug in rats (Figure 8c,d). The doses given to rats in this experiment are equivalent to the recommended human doses, as described in Materials and Methods.

## 3. Discussion

A diagram summarizing the results is presented in Figure 9. The goal of this work was to develop models of a viral infection in mice and rats and use the models to study the antiviral effects of two compounds, Tilorone and Cridanimod. Compared with mice, laboratory rats are very rarely used in models of infection with alphaviruses, due to the fact that rats probably have a high natural resistance to alphaviruses. In general, rat models of lethal viral infections are rare and require defined rat lines. Developing such models is greatly needed for pharmacovirological research, given the potential to better recapitulate effects anticipated in humans.

Tilorone and Cridanimod are traditionally referred to as IFN-inducers. Both compounds are registered drugs in the author’s country and beyond. The group of IFN-inducers existed in the national pharmacopoeias before COVID-19, but these drugs have become the prescriptions of choice since the beginning of the epidemic. This is because Tilorone and Cridanimod are available whereas specific SARS-CoV-2 antivirals are limited. Low-Mw IFN-inducers are not in clinical use as antivirals in modern Western medicine, but Tilorone obtained FDA-approval in the mid-1970s. We did not find current or finished FDA-approved clinical trials (CT) devoted to the use of Tilorone and Cridanimod as antivirals; however, Cridanimod is present in one ongoing CT for its anticancer activity. Tilorone and Cridanimod had been extensively researched in the West (review in [10]). It was reported that Tilorone was withdrawn from prescription in the USA, not because of a lack of clinical efficacy, but to avoid side effects [16]. These compounds remained as over-the-counter (OTC) drugs in countries of the former USSR, Eastern Europe, and China for a long time. Russian medical journals have published a voluminous corpus of studies on using the antiviral action of Tilorone or Cridanimod for both treatment and prophylaxis (partially reviewed in [17]). Recent examples are [18,19].

Tilorone and Cridanimod had been developed long ago in the USA in the search for small molecules to be able to induce IFNs in mice [10]. As is expected, both Tilorone and Cridanimod show antiviral activity in mice [6]; however, it appeared that these drugs do not induce circulating IFNs in larger animal species and humans. No induction was found in rats, at least in dosages that are equivalent to the doses recommended for humans [8]. A mechanism of the induction of IFN-α/β, which was deciphered at the molecular level in mice, will not work in humans because the molecular target, STING, differs between the species [8]. In previous research, Tilorone induced the low-level IFN-like activity (IFN subtypes were not determined) in cultured human cells only when high, nearly cytotoxic, doses were used. Moreover, the inducing effect was observed only in specific cell types of the immune origin. Finally, because the antiviral effect in mice has been attributed to the IFN-inducing capacity, its absence in humans could have been interpreted as a failure of prospects for these compounds to be used as clinical drugs.

In our study, the existence of the antiviral effect of Tilorone and Cridanimod was confirmed, including the protection from lethal infection in mice (expectedly) and the ability to reduce viremia in rats (less expectedly). The doses in our study are equivalent to the recommended human therapeutic doses, and they represent the lower end of a range of doses in the literature because the majority of papers describe effects at higher dosages (100–250 mg/kg Tilorone, or >300 mg/kg Cridanimod to mice, per os) [20].

The temporal course of IFN-α/β in rats after giving low-Mw inducers has not been studied with any precision. Available literature is scant and presents only single-time-point measurements. Moreover, the available literature is old, and it describes total IFN-like activity in samples without the determination of type-I IFN subtypes. We interpret this lack of interest in using rats for testing low-Mw IFN-inducers as a probable indication of poor induction in this species.

We found that the antiviral effect of Tilorone and Cridanimod in rats is not linked to the induction of circulating IFN-α or IFN-β. Other researchers who worked with mice also did not find a correlation between the levels of the induced IFNs-like activity (no type-I IFNs subtypes were determined separately) and protection [21]. Our research and the research of others, include observations that allow us to postulate that a different IFN-independent and species-independent antiviral mechanism exists for Tilorone and Cridanimod. The different mechanism also probably functions in mice; however, it becomes evident in the absence of the induction (e.g., in experiments on rats or during treatment of human patients). It is unlikely that Tilorone or Cridanimod are direct-acting antivirals (DAA) because in different research, their activity (including the protective effect) was shown against members of unrelated viral families such as Filoviridae, Hepadnaviridae, Herpesviridae, Orthomyxoviridae, Picornaviridae, Togaviridae, Rhabdoviridae, and Flaviviridae [10]. Given the diversity of inhibited viruses, it is improbable that Tilorone or Cridanimod are capable of specifically recognizing a viral protein target, binding to it, and inhibiting a specific viral component, as selective inhibition is a defining feature of DAA.

There are confirmations from different experimental setups that Tilorone has the broad-spectrum antiviral activity without inducing type-I IFNs. For example, Tilorone inhibits Chikungunya virus (fam. Togaviridae) and Middle East Respiratory Syndrome Coronavirus (MERS-CoV) (fam. Coronaviridae) in Vero cell cultures with a low median effective concentration (EC_50_) 3–4 μM [11]. The Vero cells used in the cited study are not of murine origin, and finally, Vero cells are defective in type-I IFNs’ signaling [22]. No similar data have been published for Cridanimod.

To offer an explanation for the observations made by us and another group on the lack of a correlation between the induced IFN-α/β levels and antiviral effect, we postulate that Tilorone and Cridanimod are capable of establishing an IFNs-independent antiviral state. This capability explains the virus-suppressing activity in species other than mouse and therapeutic successes in humans. The hypothesized capacity of Tilorone and Cridanimod in inducing the IFNs-independent antiviral state will function without increasing IFN-α/β in serum in dose ranges equivalent to the human doses.

The antiviral state presents with the transcriptional activation of numerous (~100) defensive genes called IFN-stimulated genes (ISG); however, a fraction of ISG also becomes transcriptionally active during infection, even in the absence of the extracellular IFNs-mediated signal because of the redundancy in the regulation [23]. The IFNs-independent antiviral state is also possible [23]. Currently, there is insufficient evidence to draw a map of biochemical pathways connecting intracellular Tilorone- or Cridanimod-targets to the IFNs-independent antiviral state. Nevertheless, we can mention the existence of a cross-talk between cellular stress signaling, expression of genes controlling cellular adaptation to stress, and the antiviral state [24]. Tilorone and Cridanimod show biochemical interactions, in addition to engaging murine STING, whose interactions also work in non-murine systems, and thus, are supposed to be species-independent. Such functions are inducing cellular hypoxia, activating hypoxia-response genes, and stabilizing cytoplasmic DNA [7,16]. There are indications that Tilorone and Cridanimod are inhibitors of mitochondrial breath and that they induce apoptosis through the mitochondrial pathway [25]. It has become a novel field of research that intracellular hypoxic stress triggers homeostatic pathways, which, in turn, have crossroads with antiviral defense pathways [24].

Tilorone induces “IFN-related DNA damage resistance signature” (IRDS) [26], which is a clinical term describing an increased expression of seven IRDS genes: STAT1, MX1, ISG15, OAS1, IFIT1, IFIT3, and IFI44 [27]. With regard to this capacity, five of these ISGs (ISG15, OAS1, IFIT1, IFIT3, and IFI44) are the same genes which participate in the IFNs-independent antiviral state [23]. Exact cell populations in the body, which are primarily targeted by Tilorone or Cridanimod to exert their antiviral effect, are not known. One study focuses on PBMCs [25].

VEEV is amenable to easy genetic manipulations due to the small genome size and availability of full-genome clones. The main molecular virulence determinants for the VEEV subtype IAB (to which TC-83 and TrD belong) are the third position in the genomic region 5′UTR (A in TC-83, G in TrD) and the 120th amino acid residue in E2 (Arg in TC-83, Tre in TrD) [15].

One novelty in this study is that a rat model of viral infection was created using the engineered virus to complement the unsolved deficit of available rat models for pharmacovirology. The feasibility of the created model for the antiviral action of Tilorone and Cridanimod was demonstrated in this study. The viruses in this study were produced from molecularly cloned genomes, and thus, are genetically pure because we did not use infectious passages to propagate viruses. The three viruses in this work show different viremia dynamics in mice and rats. Only one strain, cTC-83/TrD, which is the closest to the wild-type prototype was virulent for adult mice among three tested VEEV strains. This strain differs from the parental vaccine strain TC-83 by only two nucleotide substitutions in the viral genome, one substitution in 5′-UTR, and one in E2. The substitutions locate the major virulence determinants [15]. Thus, only two nucleotide substitutions are sufficient to reverse the avirulent phenotype to a virulent phenotype; however, the most virulent strain, cTC-83/TrD, does not kill adult rats, at least when rats were infected with 100 PFU. In the published literature, there is just one paper with data on mortality in laboratory rats from VEEV [28]. The mortality in rats caused by the passaged (not molecularly cloned) strain TrD was 50% when the rats were infected with a very high dose 10^6^ PFU [28]. We were unable to cause the lethal infection in rats in different experiments wherein rats were infected with doses up to 10^6^ PFU cTC-83/TrD (results not shown). It is likely that changing only two major virulence determinants in the context of the attenuated vaccine backbone is insufficient to restore the full virulence of the wild type.

With regard to the published prototypic virus genomes, the cTC-83 genome is the most similar to the pVE/IC-92 virus from [29]. Our cTC-83/TrD virus resembles the biological properties that the V3000 virus described in [30]. The median survival time for cTC-83/TrD is indistinguishable from that of the wild-type strain, TrD; however, LD_50_ for cTC-83/TrD is about 50 times higher (SC LD_50_ for adult mice: 10 PFU for cTC-83/TrD; 0.2 PFU for the original TrD strain).

An unexpected observation in this study is that the virus cTC-83/TrD-GFP, which carries two copies of SP and a GFP insert, appears to be attenuated and avirulent to adult mice. It is unlikely that the insertion of GFP gene itself results in this pronounced attenuation. We hypothesize that the attenuation is caused by molecular interactions in the cTC-83/TrD-GFP genome which are unfavorable to virus replication, specifically by a phenomenon of interference between different copies of SP which are present in the genome.

It has become apparent during the SARS-CoV-2 pandemic that epidemiological well-being is fragile. Healthcare must be prepared to combat emerging pathogens for which there are no specific treatments beforehand. Tilorone and Cridanimod have proven their systemic curative effect during infections with a broad range of viruses. Thus, Tilorone and Cridanimod must be considered as the potential prophylaxis at the beginning of an emerging epidemic.

The discovery of more chemical structures with the ability to induce IFN-α/β in humans is needed and justified by the clinical perspective. In addition, as exemplified by Tilorone and Cridanimod, some compounds appear to be broad-spectrum antivirals, even in the absence of circulating IFNs. This requires further research on the inducible IFNs-independent antiviral state.

## 4. Materials and Methods

### 4.1. Genetic Engineering

Molecular infectious clones (MICs) are plasmids containing full-length cDNA of the VEEV genome under the control of eukaryotic transcriptional control elements, allowing the rescue of the live virus using DNA-transfection. In our MICs, the VEEV genome is cloned downstream of the human cytomegalovirus (CMV) immediate-early promoter. The poly-A sequence, which is present at the 3′-terminus in the natural VEEV genome is partially preserved in the form of a 26-nt-long oligo-adenine sequence. The antigenomic ribozyme of hepatitis D virus and polyadenylation signal of the human growth hormone gene are cloned following the oligo-A. More details about the genetic organization of the VEEV MICs may be found in the previous paper [31]. An initial plasmid used to develop the MICs has been published under the name pCMV-VEE-GFP [31].

The MIC cTC-83 (Figure 1) carries the genome of the vaccine strain TC-83 (the first letter in the MIC’s name stands for “cloned”). The cloned viral genome was sequenced at full length, and it has the same sequence as published in Genbank (accession no. EEVNSPEPB).

The cTC-83/TrD virus was produced from cTC-83 by making two nucleotide substitutions, one in the 5′-untranslated region (5′UTR) and one in the E2 gene. The third nucleotide in 5′UTR is guanine in cTC-83/TrD versus adenine in cTC-83. The substitution in the E2 gene is G359->C (numbering is local, and pertains to a particular gene or protein). The encoded amino acid residue in E2 position 120 changes from arginine (cTC-83) to treonine (cTC-83/TrD).

The virus cTC-83/TrD-GFP was produced from cTC-83/TrD by inserting a GFP gene in the viral genome. The 43-nt—long intergenic region, which contains a viral promoter for 26S subgenomic RNA (SP), was duplicated and the GFP gene was cloned under the control of the first SP copy. The viral genes, C-E3-E2-6k-E1, encoding structural proteins, remain under the control of the second SP copy (Figure 1).

Nucleotide sequences of the MICs, cTC-83, cTC-83/TrD, and cTC-83/TrD-GFP, are listed in Appendix A.

### 4.2. Cell Line and Production of Viruses

Baby hamster kidney cells (BHK-21, ATCC CCL-10) were grown in DMEM with high glucose (Lonza Cat# BE12-604, Morristown, NJ, USA), containing 10% fetal bovine serum (FBS, Gibco Cat# 16000-044, Amsterdam, The Netherlands), 2 mM L-glutamine, 1% MEM vitamin solution (ThermoScientific Cat# 11120052, Gloucester, UK), 1% non-essential amino acids (ThermoScientific Cat# 11140050, UK), 100 U/mL penicillin, and 100 μg/mL streptomycin.

The viruses cTC-83, cTC-83/TrD and cTC-83/TrD-GFP were rescued from the MICs using DNA transfection. Lipofectamine 2000 (ThermoScientific Cat# 11668019, Pleasanton, CA, USA) was used for transfections. BHK-21 cells were grown in 6-well plates to 70% confluence. Plasmid (MIC) DNA was taken in the amount of 2 μg per well and mixed with 30 μL OptiMEM I (ThermoScientific Cat# 11058021, CA, USA) without antibiotics and FBS. The Lipofectamine 2000 reagent was mixed with 90 μL OptiMEM I in a different tube. The DNA and Lipofectamine solutions were combined, vortexed, and incubated at room temperature for 20 min. Then, media were removed from the wells, the cells were rinsed with phosphate buffered saline (PBS, ThermoScientific Cat# 14190144, USA), and covered with the transfection mixture. The serum-free OptiMEM I medium was added to the transfected cells (2 mL per well) and the plates were incubated for 18–20 h. The next day, media were replaced with the complete medium (DMEM with 10% FBS and other additives). Virus-containing media were collected at 72 hpt. Large-scale transfections were used to obtain the viruses in amounts that were sufficient for experiments on animals. For the large-scale transfections, BHK-21 cells were grown in T75 flasks (ThermoScientific Cat# 159910, USA) and transfected using proportionally increased amounts of DNA and Lipofectamine. Working with the viruses cTC-83/TrD and cTC-83/TrD-GFP, which carry the wild-type virulence determinants, transfections were conducted in the BSLIII facility in the authors’ institution.

### 4.3. Plaque Assay

Viral titers in serum samples were determined by a plaque assay on monolayers of BHK-21 cells grown in 6-well plates. Ten-fold serial dilutions of sera were prepared in PBS containing 1% FBS. The dilutions were transferred in an amount of 200 μL per well onto the BHK-21 cells. The plates were incubated for 1 h with occasional shaking. Infectious inocula were removed and the cells were covered with a molten agar-containing medium, which is the complete medium with 2% FBS and 0.6% (*w*/*v*) low-gelling temperature agarose (ThermoScientific Cat# 16520100, USA). Upon solidification of the agarose, the plates were returned to incubation for 72 h. Then, the cells were fixed in 4% formaldehyde. Plaques were stained using crystal violet. For the GFP-producing virus cTC-83/TrD-GFP, both GFP-positive foci and plaques visible in stained monolayers were counted.

### 4.4. Animals

The study was approved by the Institutional Animal Care and Use Committee (IACUC) of the National center for biotechnology (decision 02/AP08855853 dated 6 October 2020). The reporting in this study is in accordance with guidelines published by the National Centre for the Replacement, Refinement and Reduction of Animals in Research (ARRIVE guidelines). All research work with laboratory animals was performed in accordance with generally accepted ethical standards for the treatment of animals based on standard operating procedures that comply with the rules adopted by the European Convention for the Protection of Vertebrate Animals Used for Research and Other Scientific Purposes.

Adult male outbred CD-1 mice 8–10 weeks of age, weighing 24 ± 2 g, and adult male Wistar rats (outbred albino rats) aged 10–12 weeks, weighing 200–300 g, were used in experiments. These lines of rodents are bred in the animal facility of the National Center for Biotechnology, Nur-Sultan, Kazakhstan. The animals were housed in a room with a controlled temperature and a 12 h light-dark cycle with unlimited access to standard food (SSNIFF V1534-300, HTLab AG, Heideck, Germany) and drinking water ad libitum.

Two weeks before experiments, animals were randomly distributed in cages by ten mice per cage or three rats per cage. The animals were used after a 14 day adaptation period. The animals were preserved in the same groups and same cages to reduce the stress.

As required by the IACUC for survival experiments, moribund animals meeting the euthanasia criteria were euthanized using CO_2_. The euthanized animals were counted as dead.

### 4.5. Antivirals

Tilorone (2,7-bis[2-(diethylamino)ethoxy]fluoren-9-one hydrochloride) and Cridanimod (10-carboxymethyl-9-acridanone, CMA), the structural formulas of which are presented in the Appendix A, are both registered drugs in the authors’ country. The drugs were purchased in a pharmacy as tablets containing 125 mg of the active substance per tablet. Tilorone was produced by JSC Nizhfarm, Russia. Cridanimod was produced by LLC Polisan NTFF, Russia (brand name “Cycloferon”, this is a CMA salt containing N-methylglucamine as a counter-ion). The drugs were administered intragastrically (IG) using tablets dispersed in a starch paste. Animal doses per kg weight were calculated by taking pharmacopoeia-recommended doses for adult humans (2 mg/kg for Tilorone, 10 mg/kg for Cridanimod) and converting them into doses for laboratory animals by multiplying them by sensitivity coefficients (12.3 for mice, 6.2 for rats) to account for differences in the metabolism intensity [32]. Hence, IG doses for mice were 25.6 mg/kg Tilorone and 123 mg/kg Cridanimod; for rats, doses were 13 mg/kg Tilorone and 62 mg/kg Cridanimod.

### 4.6. Experimental Designs

Experiments on animals were repeated twice or thrice as described in the Results. To study the viremia in untreated mice, three groups of mice (*n* = 18 per group) were injected subcutaneously (SC) with 10^3^ PFU of cTC-83, cTC-83/TrD, or cTC-83/TrD-GFP. Three mice from each group were euthanized at time points 8, 24, 48, 72, 96, and 120 hpi. Blood was collected from the heart and used to produce serum. Sera were stored at −80 °C until the plaque assay was performed. The same experimental design was used to study viremia in untreated rats.

The effect of the drugs on the viremia in mice was studied using the viruses which demonstrated sufficient viremia, cTC-83/TrD and cTC-83/TrD-GFP. Three groups of mice (*n* = 18) were used for each virus. Cured groups received the drugs IG at the beginning of the experiment and in 24 h intervals until the completion of the experiment. The dosing is described in the “Antivirals” section. A control group received a placebo (starch paste) using the IG method and same regimen. The animals were infected by SC injection with 10^3^ PFU immediately after receiving the first dose of a substance. Three mice per group were terminated at 4, 8, 24, 36, 48, and 72 hpi to obtain blood sera.

Experiments to measure drugs effects on viremia in rats were planned similarly with the same infectious doses and timing, but using rats’ dosing of Tilorone and Cridanimod.

The median lethal dose (LD_50_) in mice for the virus cTC-83/TrD was determined as follows. Ten-fold serial dilutions of a cTC-83/TrD stock were prepared. Ten groups of mice (*n* = 10) were injected SC with the dilutions (from 1:10 to 1:10^10^). The virus inoculum dose was 100 μL per animal. The mice were monitored for 10 days for death or terminal disease. The Reed–Muench method was used to calculate a virus titer in LD_50_ units. The titer in plaque-forming units (PFU) was determined in the virus stock and one LD_50_ was expressed in PFU.

An experiment to measure protection in mice from death caused by the virus cTC-83/TrD was as follows. Three groups of mice (*n* = 10) were injected SC using 100 PFU per animal. The mice received the first dose (IG) of Tilorone, Cridanimod, or a placebo immediately after being injected with the virus. The same substances were given each day for ten days. The mice were observed twice a day for death or terminal disease.

### 4.7. Measuring Induced Interferon

Induced IFN-α and IFN-β were measured during a course of treatment with daily IG doses of Tilorone or Cridanimod in mice and rats. Three groups of animals (*n* = 18) were used. The animals received the drugs or a placebo at the beginning of the experiment and at 24 and 48 h. Doses for mice were 25.6 mg/kg Tilorone and 123 mg/kg Cridanimod (i.e., the same as used in experiments to measure the effect on viremia). The doses for rats were 13 mg/kg Tilorone and 62 mg/kg Cridanimod. Three animals from each group were terminated to collect sera at time points 4, 8, 24, 36, 48, and 72 hpi. IFN-subtype-specific commercial ELISA kits were used to measure IFN-α in mice (MyBioSource MBS260421, San Diego, CA, USA) or rats (MyBioSource MBS267050, CA, USA), as well as kits for IFN-β in mice (MyBioSource MBS701008, CA, USA) and rats (MyBioSource MBS703132, CA, USA).

### 4.8. Statistical Comparisons

Data were analyzed using GraphPad Prism software v.8.4.2 (GraphPad Software LLC, San Diego, CA, USA). Viral titers were log-transformed, and the unpaired t-test was used to compare titers between groups. The log-rank Mantel–Cox test was used to compare differences between the numbers of survivors. All tests were two-sided, and differences were considered statistically significant if the *p*-values were less than 0.05.

## 5. Conclusions

In this study, we measured the antiviral activity of synthetic chemical compounds, Tilorone and Cridanimod, which hold promise as clinical antivirals, despite the fact that their mechanism of action in humans is still a subject of controversy. Small animal models of viral infection, utilizing engineered strains of the alphavirus Venezuelan equine encephalitis virus, are described. Tilorone and Cridanimod has proven a systemic antiviral effect in rats, whose models profile similarly to the human response. These compounds deserve inclusion in the appropriate clinical trials as antivirals.

## Figures and Tables

**Figure 1 pharmaceuticals-15-00617-f001:**
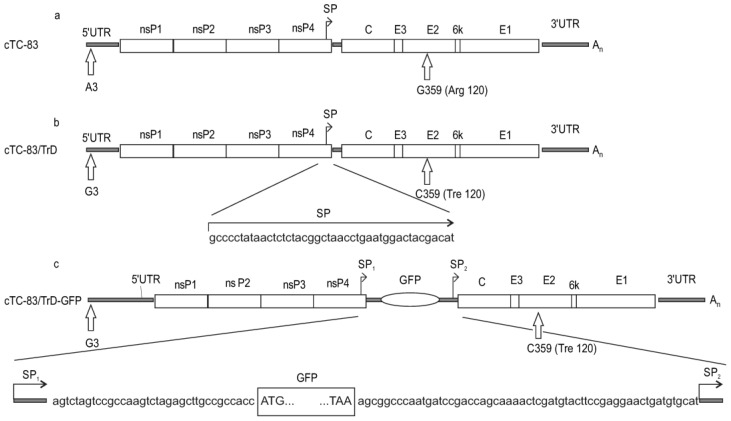
Schematics of genomes of variants of the Venezuelan equine encephalitis virus (VEEV) created in this work: cTC-83, cTC-83/TrD, and cTC-83/TrD-GFP. Viral genes are shown as open boxes. Arrows indicate locations of the viral subgenomic promoter (SP). Moreover, 5′UTR and 3′UTR are untranslated regions. nsP—nonstructural proteins genes. C-E3-E2-6k-E1—viral genes for structural proteins. A polyA tail at the genomic 3′-terminus is designated as A_n_. Prefix “c” in the viruses’ names stands for “Cloned”. Panel (**a**), cTC83 is the molecularly cloned vaccine strain TC-83. Panel (**b**), cTC-83/TrD differs from cTC-83 by single-nucleotide substitutions which locate to the major virulence determinants, namely, in 5′UTR (G3) and E2 (C359). The encoded amino acid residue in E2 position 120 is treonine in cTC-83/TrD or arginine in cTC-83. The presence of guanine in position 3 (5′UTR) and cytosine in position 359 (E2) bring the cTC-83/TrD genome closer to the virulent strain Trinidad donkey (TrD). In the panel (**b**), 43-nt—long sequence of the VEEV SP is shown beneath the arrow. Panel (**c**), TC-83/TrD-GFP carries two SP copies and a GFP gene. The GFP gene is cloned under the control of the first SP copy. Viral genes C-E3-E2-6k-E1 remain under the control of the second SP copy. A modified genomic region with the GFP gene is shown at the bottom of panel (**c**). The GFP gene starting and terminating codons are enclosed in the rectangle.

**Figure 2 pharmaceuticals-15-00617-f002:**
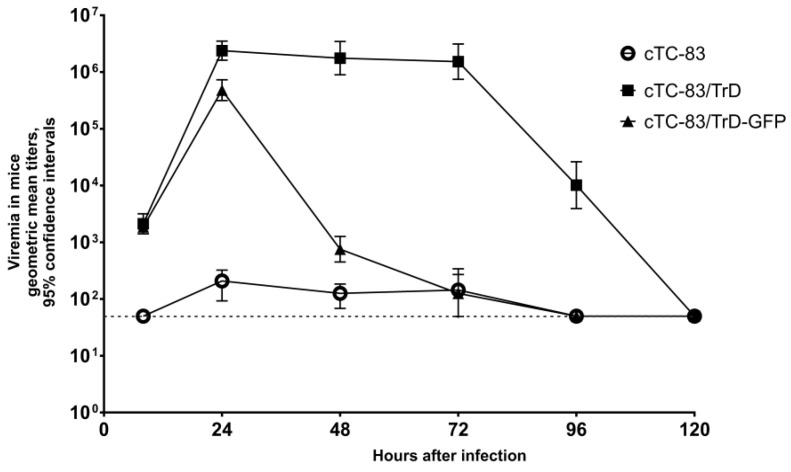
Viremia in mice. Mice were injected SC with 1000 PFU. Geometric mean titers from three mice per point with 95% confidence intervals (CI) are shown. The horizontal dashed line indicates the limit of detection. Two replicates of this experiment produced similar results. The results of one experiment are shown.

**Figure 3 pharmaceuticals-15-00617-f003:**
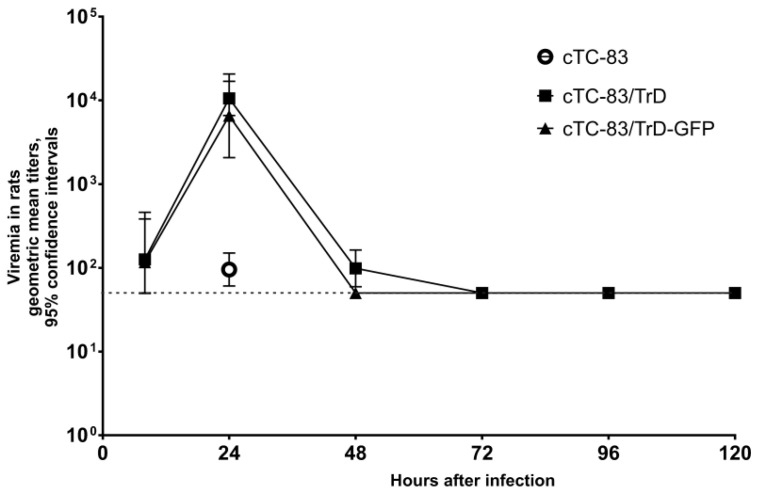
Viremia in rats. Rats were injected SC with 1000 PFU. Geometric mean titers with 95% CI are shown. For cTC-83, the only time point when the virus was detected in rats’ sera was 24 hpi, and samples collected at other times were negative in plaque assay. The horizontal dashed line is the limit of detection. The experiment was repeated twice with similar results. The results for one replicate are shown.

**Figure 4 pharmaceuticals-15-00617-f004:**
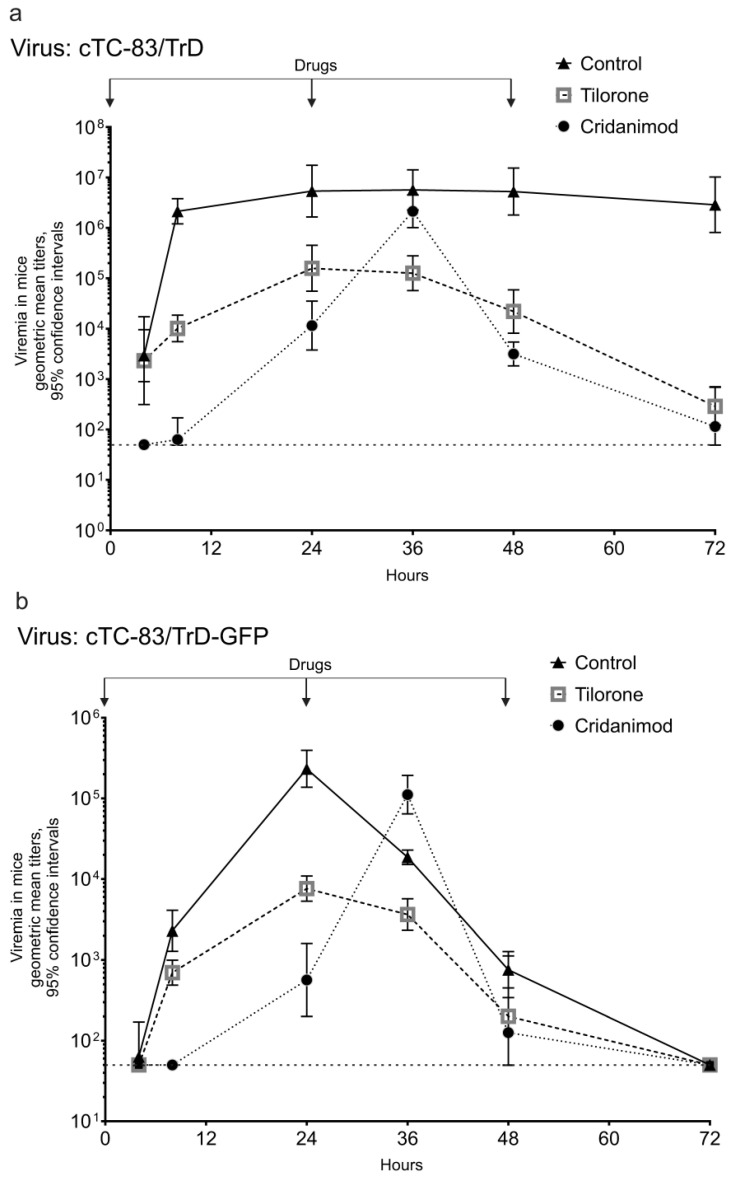
Effects of Tilorone and Cridanimod on viremia in mice. Mice were given IG drugs or a placebo at 0, 24, and 48 hpi (times of administration are indicated with arrows). Immediately after the first administration, the mice were injected SC with 1000 PFU. Geometric mean titers from three mice per point with 95% CI are shown. The horizontal dashed line is the limit of detection. This experiment was repeated three times with quantitatively and qualitatively similar results. (**a**) Data for mice infected with cTC-83/TrD. (**b**) Mice were infected with cTC-83/TrD-GFP.

**Figure 5 pharmaceuticals-15-00617-f005:**
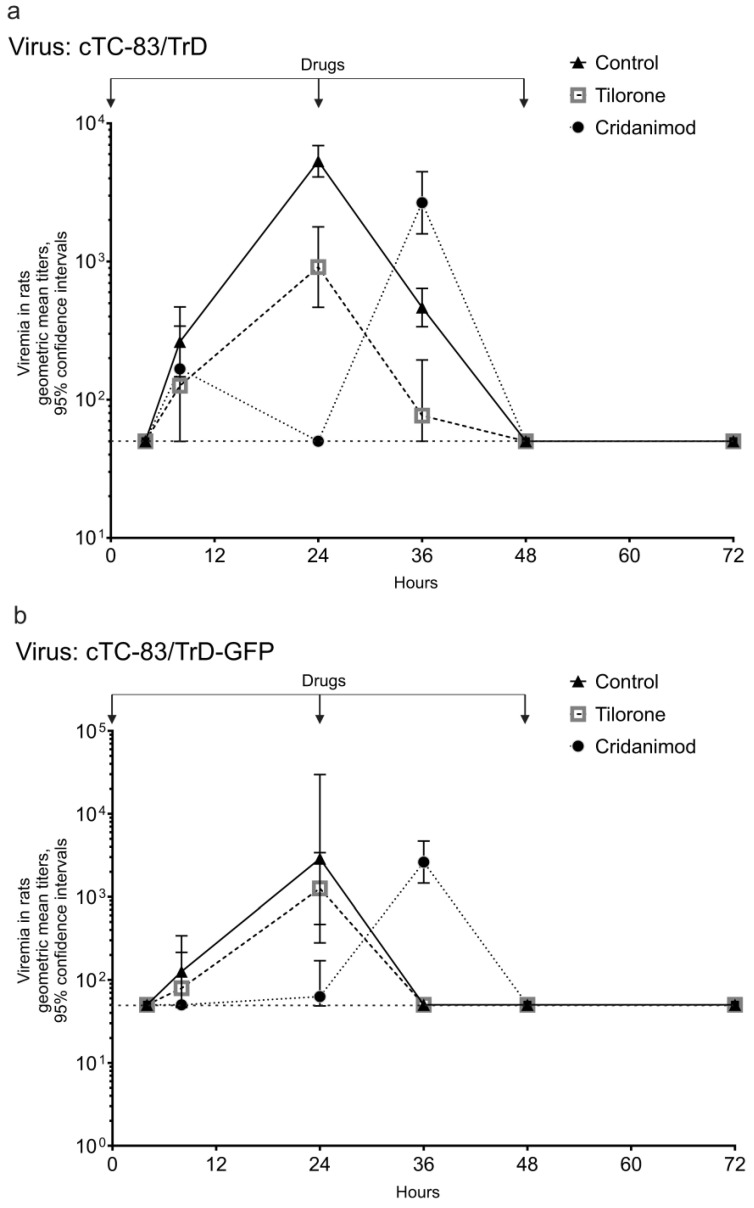
Effects of Tilorone and Cridanimod on viremia in rats. Rats were given IG drugs or a placebo at 0, 24, and 48 hpi. Immediately after the first dose, the rats were injected SC with 1000 PFU. Geometric mean titers with 95% CI are shown. The horizontal dashed line is the limit of detection. Results from one of three replicate experiments are shown. (**a**) Viremia for cTC-83/TrD. (**b**) Viremia for cTC-83/TrD-GFP.

**Figure 6 pharmaceuticals-15-00617-f006:**
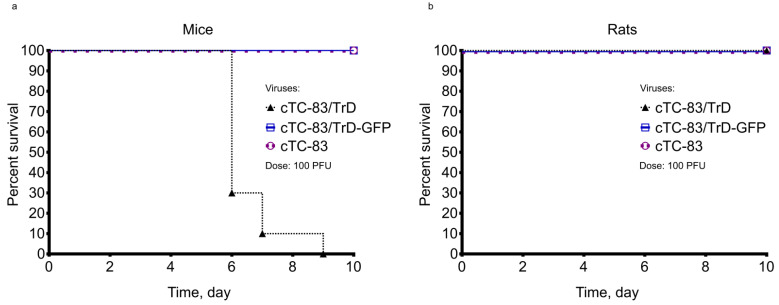
Survival in animals infected with viruses cTC-83 (vaccine-derived), cTC-83/TrD (same as the virulent strain TrD by the major virulence determinants), and cTC-83/TrD-GFP (expresses GFP during intracellular replication, has two copies of SP in the genome). (**a**) Survival in mice. Only cTC-83/TrD was virulent with 100% morbidity and mortality. (**b**) One-hundred percent survival was recorded in rats infected with either virus.

**Figure 7 pharmaceuticals-15-00617-f007:**
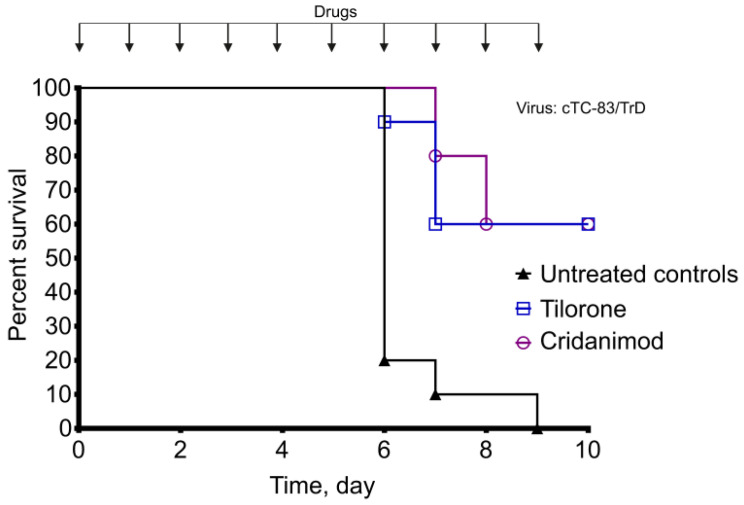
Tilorone and Cridanimod protected mice from lethal infection. Three groups of mice (*n* = 10) received IG administrations of Tilorone, Cridanimod, or a placebo at the beginning of the experiment and every day until the completion. The mice were injected SC with 10 LD_50_ of the virulent cTC-83/TrD virus at day zero. This experiment was performed twice. Results from one experiment are shown.

**Figure 8 pharmaceuticals-15-00617-f008:**
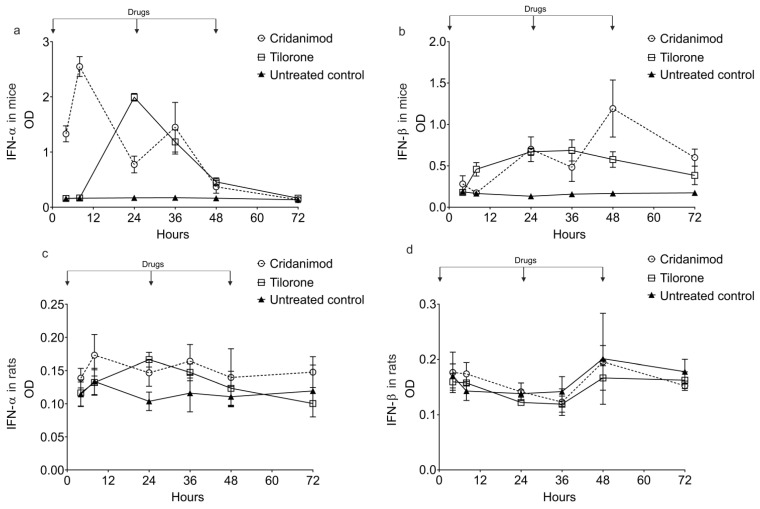
Animals received Tilorone, Cridanimod, or a placebo. The substances were given at 0, 24, and 48 h. Levels of IFN-α and IFN-β in sera were determined with ELISA. Optical densities (OD) are shown. Y-axes show the mean OD for three animals per point with standard deviations. (**a**) Levels of IFN-α in mice. (**b**) IFN-β in mice. (**c**) IFN-α in rats. (**d**) IFN-β in rats.

**Figure 9 pharmaceuticals-15-00617-f009:**
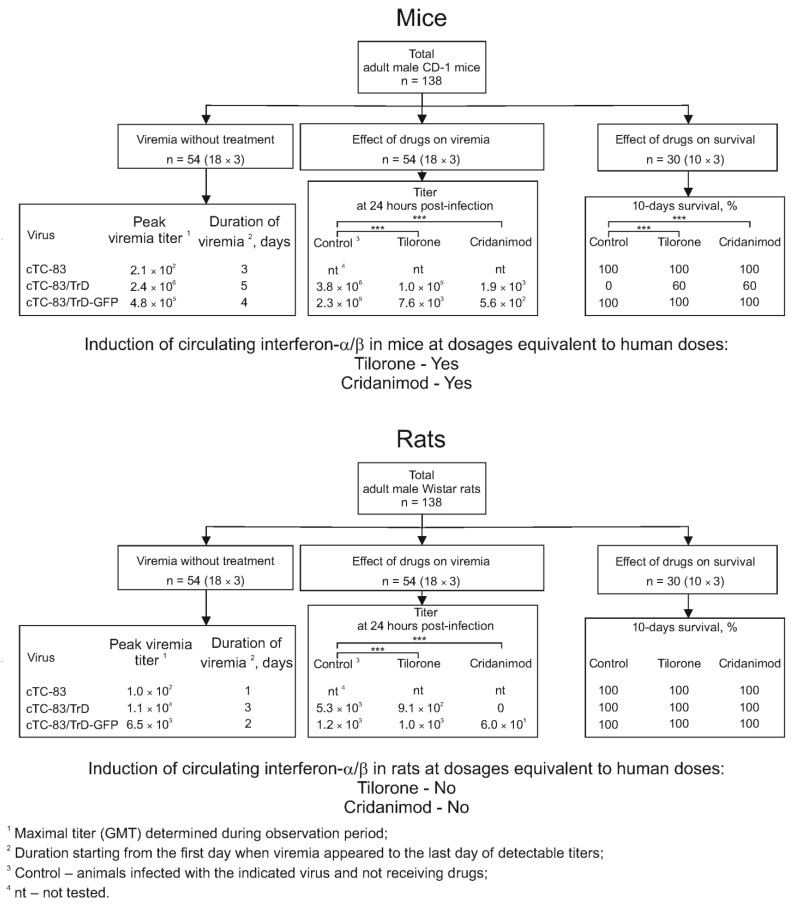
Summary diagram for the results. Statistical significance markings are shown only for the most virulent virus cTC83/TrD to preserve picture clarity: ***, *p* < 0.001.

## Data Availability

Data is contained within the article and Appendix A.

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
