# Peer review of "Tilorone and Cridanimod Protect Mice and Show Antiviral Activity in Rats despite Absence of the Interferon-Inducing Effect in Rats"

_pharmaceuticals, 2022, doi:10.3390/ph15050617_

Round 1

Reviewer 1 Report

The work presented by Keyer et al aims to test the antiviral properties of two compounds, Tilorone and Cridanimod, known as antivirals, on mice and rats, previously infected with three recombinant viruses, cTC-83, cTC-83/TrD and cTC-83/TrD-GFP, generated by them.

They initially evaluated the ability of this viruses to induce viremia in both mice and rats. The results presented show that the cTC-83/TrD and the cTC-83/TrD-GFP viruses induce an increase in virulence compared to cTc-83 of about 3/4 times in mice, and 2 times in rat after 24h. While all three kinds of viruses induce high levels of viremia in both animals, it rapidly drops in rats within 48h after infection. The Tilorone and Cridanimod treatment affect viremia in mice but seems do not have the expected results in rat models. Then they test this compounds on IFN induction. And even in this case the expected results don’t be confirmed. So, the final question is: Did the authors bring anything new to the scientific world? In my opinion no.

They knew that Tilorone and Cridanimod don’t increase the INF levels, but they dose only that one, why didn't they look to other inflammatory mediators for viral infection?

They said that their goal was to develop modes of viral infection in rat, but the recombinant viruses cTC-83/TrD and cTC-83/TrD-GFP give a viremia in rat of 24h that spontaneously degrees (fig 3), in fact no rats died after infection (fig6).

There are two supplemental figures without control cell, I mean cell not infected (S7-8) and cell stained with only nuclear stain (S6).

A lot of sentences are left pending, and the alpha and beta symbols are lost most of the time IFN is mentioned in the text.

Conclusions are lack.

Author Response

1) Reviewer’s comments: >> They initially evaluated the ability of this viruses to induce viremia in both mice and rats. The results presented show that the cTC-83/TrD and the cTC-83/TrD-GFP viruses induce an increase in virulence compared to cTc-83 of about 3/4 times in mice, and 2 times in rat after 24h. While all three kinds of viruses induce high levels of viremia in both animals, it rapidly drops in rats within 48h after infection. The Tilorone and Cridanimod treatment affect viremia in mice but seems do not have the expected results in rat models. Then they test this compounds on IFN induction. And even in this case the expected results don’t be confirmed. So, the final question is: Did the authors bring anything new to the scientific world? In my opinion no.

 - Answers

The developed rat model produces reproducible results and the results show that Tilorone and Cridanimod are efficient antivirals. Despite the viremia in rats is not long-term, it is reproducible and the titers allow measuring differences in the viremia inflicted by drugs. In fact, developing rat models to directly test drugs influence on viremia is a complex problem not solved by other groups.

-           Our described model allowed us to collect data for the proof of the antiviral action of two drugs in the absence of interferon induction. The action of Tilorone and Cridanimod in rats is not just statistically significant but bright (Figure 5a, data for the virulent virus). This is an illustration of the feasibility of the rat model, and not many groups in the world have the access to similar models.

-           Our results are unique because there are no other published papers on the influence of IFN-inducers in rats on the course of viremia.

2) >>They knew that Tilorone and Cridanimod don’t increase the INF levels, but they dose only that one, why didn't they look to other inflammatory mediators for viral infection?

 - Answers:

  1. The mechanism of action of Tilorone and Cridanimod in mice is known and was proven to include the action of type-I IFNs. Then the first we needed to check is if type-I IFNs are induced in other species.
  2. The main controversy about Tilorone and Cridanimod is that they are dubbed IFN-inducers. The medical community in our country and beyond believes that these drugs induce interferons. It was necessary to check exactly is the induction of interferons (not numerous other cytokines) is the cause why the compounds have the antiviral effect.

3) >> They said that their goal was to develop modes of viral infection in rat, but the recombinant viruses cTC-83/TrD and cTC-83/TrD-GFP give a viremia in rat of 24h that spontaneously degrees (fig 3), in fact no rats died after infection (fig6).

- Answers:

Developing models of viral infection in rats feasible to testing antivirals is a complex task which is still not solved entirely. In fact, we demonstrated the reproducible viremia in rats and our results show not only statistically significant but bright effect of the drugs in our rat model. This proves the feasibility of our created model in rats for further research on Tilorone and Cridanimod. Figure 9 presents summary data and the statistical significance of the antiviral effect in the model with the virulent virus and rats.

4) >> There are two supplemental figures without control cell, I mean cell not infected (S7-8) and cell stained with only nuclear stain (S6).

- Answers:

Corrected. New photographs are added to the Supplementary Information depicting control uninfected cells (Figure S5ab). For the cells infected with the GFP-producing virus double-images (in white light and GFP-fluorescence) are shown in Figures S5ab and S6ab.

5) >> A lot of sentences are left pending, and the alpha and beta symbols are lost most of the time IFN is mentioned in the text.

- Answers:

The manuscript has passed through English-language editing service available at the authors’ organization. This was done after receiving the reviewers’ comments.

6) >> Alpha and beta symbols were placed at IFN throughout the manuscript at the appropriate context.

- Answers:

Please note, when we reference other groups data on determining IFN in rats we refer to the “IFN-like activity” without mentioning IFN-a/b because the cited papers do not report particular type-I IFN subtype.

6) >> Conclusions are lack.

- Answers:

Corrected. New conclusion is in lines 612-619 in the revised manuscript.

Reviewer 2 Report

 Tilorone and Cridanimod protect mice and show antiviral activity in rats despite absence of the interferon-inducing effect in  rats

The results of the study show that interferon independent and species independent antiviral mechanism is induced by Tilorone and Cridanimod. This shows that further studies can be undertaken to assess  these drugs for the treatment of viral diseases affecting human beings and animals.  The article is suitable for publication. Corrections suggested below may be carried out before final submission.

Corrections/modifications required

Section

Line No.

Corrections

Results

152

All viruses become .......at 72 hpi. 

This sentence has to be corrected because the figure 2 shows that cTC-83/TrD is having higher titre while the other two viruses have low titre at 72 hpi.

245-246

However all three .......sixth day.

This sentence may be rephrased.

290 and other places

IFN subtypes to be mentioned.

Discussion

383

106 PFU   to be corrected as    106PFU

Materials and methods

573

trice  to be corrected as   thrice

575 and 586

103 PFU  to be corrected as   103PFU

591 and other places

LD50  to be corrected as   LD50

   Note: Corrected/added words are underlined.

Author Response

All reviewer-suggested corrections were included in the revised manuscript.

Reviewer 3 Report

Tilorone and Cridanimod are low-molecular-weight registered antivirals in some Eurasian countries. Manuscript elaborates on the study of these drugs on antiviral action. Authors have developed the models of viral infections in mice and rats for the antiviral activity of Tilorone and Cridanimod and reported that the strains have different virulence having a strain with   molecularly cloned attenuated vaccine and another  strain is having the major virulence determinants like virulent strains. The third virus has wild-type virulence determinants and in addition is engineered to express GFP during the replication. Tilorone and Cridanimod were used to treat experimentally infected animals and the used dosage was equivalent to the human treatment protocol suggesting that  they have antiviral effects and protection. The antiviral effect is not linked to the induction of interferon-alpha or interferon-beta in rats. 

    Experiments are nicely planned and obtained results are discussed with appropriate citations. However, I would suggest to get the language checked for punctuations and spell checks. I recommend the manuscript for publication with my above observations.

Author Response

Thank you for your kind opinion!

The manuscript has passed through English-language editing service available at the authors’ organization. This was done after receiving the reviewers’ comments.

The language was checked to correct the style, punctuations and spelling.

Reviewer 4 Report

This work reported that Tilorone and Cridanimod protect mice and show antiviral activity in rats despite absence of the interferon-inducing effect in rats. There are some issues in this manuscript that should be addressed as follows:

  • Author names: There is no need to add “1” after the author names because all of them belong to the same institution.
  • Abstract:

- The strain of the used mice and rats should be mentioned.

- The meaning of the abbreviations should be clearly defined at their first mention (e.g. GFP).

- Key words: Too much key words were written. They can be reduced to 5-6 key words.

  • Materials and methods:
  1. The code of approval of the research ethics committee should be mentioned, not only the date of approval.
  2. The country to which belongs the company that supplied the used kits and chemicals should be mentioned.
  3. The scientific name of the strain of rats used in this study should be mentioned.
  4. How did you know that the animals were acclimatized?
  5. Some details about the housing conditions of the animals used in this study should be added.
  6. References for the used doses of Tilorone and Cridaminod and duration of treatment should be added.
  7. Statistical analysis: The version of the GraphPad Prism software should be mentioned.
  • Results: A collective diagram summarizing the main findings of this study is recommended.
  • Conclusion:

I think that the conclusion is not sufficient. The possible clinical implications of the results of the present study should be added.

  • References:
  1. The order of references in the text is misleading. Although “Materials and methods” section was located in the manuscript after the “Discussion” section, its references were mentioned in the references list before those of the “Discussion” section. Please, revise.
  2. Some references should be updated (e.g. Ref. 13, 14, 15)

Author Response

Reviewer’s comments

1) >> Author names: There is no need to add “1” after the author names because all of them belong to the same institution.

 - Answers:

Corrected.

2) >> Abstract:

- The strain of the used mice and rats should be mentioned.

- The meaning of the abbreviations should be clearly defined at their first mention (e.g. GFP).

- Answers:

Both points are corrected. The abstract was reworked.

3) >> - Key words: Too much key words were written. They can be reduced to 5-6 key words.

- Answers:

Done as suggested. Reduced to 6 keywords.

4) >> Materials and methods:

The code of approval of the research ethics committee should be mentioned, not only the date of approval.

The country to which belongs the company that supplied the used kits and chemicals should be mentioned.

The scientific name of the strain of rats used in this study should be mentioned.

- Answers:

All points are corrected. The abstract was reworked.

The code of approval is in lines 531-533 in the revised manuscript.

Countries of companies included. to which

The scientific name of the strain of rats is mentioned in line 541.

5) >> How did you know that the animals were acclimatized?

Some details about the housing conditions of the animals used in this study should be added.

- Answers:

These both comments are detailed in lines 542-548. All animals are bred locally. The animals were put in cages two weeks before the experiment and same groups per cage were maintained throughout experiments.

6) >> References for the used doses of Tilorone and Cridaminod and duration of treatment should be added.

- Answers:

Animal doses are equivalents to pharmacopoeia-recommended human doses. The calculations for animal doses are described in lines 560-564.

I added reference to the pharmacopoeia-recommended human doses of Tilorone and Cridaminod as well as to whole human treatment regimen. This is contemporary electronic source of drugs prescriptions in the author’s contry. Ref. 9. Lines 65 and 661.

7) >> Statistical analysis: The version of the GraphPad Prism software should be mentioned.

- Answers:

Corrected. Line 607

8) >> Results: A collective diagram summarizing the main findings of this study is recommended.

- Answers:

Created. The collective diagram summarizing the results is in Figure 9.

9) >> Conclusion: I think that the conclusion is not sufficient. The possible clinical implications of the results of the present study should be added.

- Answers:

The conclusion was reworked. The new conclusion is in lines 612-619. The clinical implications are anticipated as mentioned in lines 460 (Results) and 619 (Conclusion).

10) >> References: The order of references in the text is misleading. Although “Materials and methods” section was located in the manuscript after the “Discussion” section, its references were mentioned in the references list before those of the “Discussion” section. Please, revise.

Some references should be updated (e.g. Ref. 13, 14, 15).

- Answers:

All corrected. Whole references list was reworked. Very old references were removed to make >50% of the references within the last 5 years.

However, some selected old references cannot be sensibly replaced with newer articles because these papers present unique data, e.g. for mortality of laboratory rats from the VEEV virus. From that time no other papers were published on the same topic.

All references list was updated.

Round 2

Reviewer 1 Report

I really appreciated the effort made in trying to improve the text, which is completely varied in the spite form to the first version.
The control figures that I had asked for were added.
The diagram in Figure 9 significantly help the reader in the experimental results interpretation.